# Photovoltage-Driven Photoconductor Based on Horizontal *p*-*n*-*p* Junction

**DOI:** 10.3390/nano14181483

**Published:** 2024-09-12

**Authors:** Feng Han, Guanyu Mi, Ying Luo, Jian Lv

**Affiliations:** 1School of Defence Science & Technology, Xi’an Technological University, No.2 Xuefu Middle Road, Xi’an 710021, China; hanfeng202408@126.com; 2School of Optoelectronic Science and Engineering, University of Electronic Science and Technology of China, Chengdu 610054, China; 202011050821@std.uestc.edu.cn

**Keywords:** photodetection, photoconductor, photovoltage, *p*-*n* junction

## Abstract

The photoconductive gain theory demonstrates that the photoconductive gain is related to the ratio of carrier lifetime to carrier transit time. Theoretically, to achieve higher gain, one can either prolong the carrier lifetime or select materials with high mobility to shorten the transit time. However, the former slows the response speed of the device, while the latter increases the dark current and degrades device sensitivity. To address this challenge, a horizontal *p*-*n*-*p* junction-based photoconductor is proposed in this work. This device utilizes the n-region as the charge transport channel, with the charge transport direction perpendicular to the *p*-*n*-*p* junction. This design offers two advantages: (i) the channel is depleted by the space charge layer generated by the *p* and *n* regions, enabling the device to maintain a low dark current. (ii) The photovoltage generated in the *p*-*n* junction upon light absorption can compress the space charge layer and expand the conductive path in the *n*-region, enabling the device to achieve high gain and responsivity without relying on long carrier lifetimes. By adopting this device structure design, a balance between responsivity, dark current, and response speed is achieved, offering a new approach to designing high-performance photodetectors based on both traditional materials and emerging nanomaterials.

## 1. Introduction

A photoconductor is a classic photodetector structure with the advantages of high gain and simple configuration. According to the theoretical model of photoconductors [1], their gain (*G_PC_*) is related to the lifetime (*τ_lifetime_*) and transit time (*τ_transit_*) of photogenerated carriers, that is, *G_PC_* ∝ *τ_lifetime_*/*τ_transit_*. Therefore, the introduction of the trap effect to extend the lifetime of photogenerated carriers [2,3,4,5] or the utilization of high-mobility materials to shorten the transit time [6,7,8,9] can promote photoconductors to achieve high gain and high responsivity. However, the long lifetime of photogenerated carriers slows down the device response speed, giving rise to the gain-response speed tradeoff. Moreover, channels based on high-mobility materials like graphene typically possess high dark currents, which leads to the degradation of device sensitivity [10].

In recent years, to address the gain-response speed tradeoff, various device design schemes have been proposed, such as the use of interfacial gating [11,12], the introduction of the photovoltaic effect [13,14,15], the manipulation of the photogating effect [16,17], and the construction of an optically tunable gate [18,19]. The main concept of these schemes is to decouple the gain from the photogenerated carrier lifetime [20,21], thereby achieving high responsivity and fast response speed at the same time. However, it is still challenging to suppress dark currents while attaining remarkable responsivity and response speed. Especially for photoconductors based on narrow-bandgap semiconductors that are sensitive to infrared light, a low dark current is crucial for obtaining high sensitivity [22,23,24].

Here, a photovoltage-driven photoconductor (PVPC) is proposed to balance responsivity, response speed, and dark current. The PVPC is based on a horizontal *p*-*n*-*p* junction, in which the *n*-area is the carrier transport channel, and the two *p*-areas are located on both sides of the *n*-area. The depletion layer of the *p*-*n* junction depletes the channel and suppresses the dark current. The photovoltage that arises at the *p*-*n* junction shrinks the depletion layer and modulates the channel conductance. As a result, the PVPC simultaneously achieves a low dark current of 9 nA, a high responsivity of 19 A/W, and a fast response speed of 2.4 μs. In addition, the on-off ratio of the PVPC is improved by 277 times compared with the conventional photoconductor. More importantly, by selecting germanium (Ge), a classical narrow bandgap semiconductor, as the light absorber, the PVPC achieves efficient photodetection within the near-infrared (NIR) region.

## 2. Results and Discussion

### 2.1. Working Mechanism

Figure 1a shows the structure of the PVPC, in which carriers are transported in an *n*-type semiconductor channel. On both sides of the channel, there are two *p*-type semiconductor regions. To make the PVPC sensitive to NIR light, a narrow bandgap semiconductor material, or zero bandgap material like graphene, could be used. After charge transfer, two *p*-*n* junctions are formed in the PVPC. The depletion layers of these two *p*-*n* junctions block carriers and suppress the dark current (Figure 1b). Under illumination, a photovoltage is produced after the *p*-*n* junction separates the photogenerated electron–hole pairs, similar to the formation of open-circuit voltage in solar cells [25,26]. This photovoltage compresses the depletion layer, allowing a large number of carriers to pass through the channel and form a large photocurrent (Figure 1c). Therefore, the PVPC achieves low dark current and high photoresponse at the same time. In addition, since the photocurrent of the PVPC mainly depends on the modulation effect of the photovoltage rather than the photogenerated carrier lifetime, the PVPC can also realize fast speed.

### 2.2. Basic Properties

The doping concentration profile of the PVPC is shown in Figure 2a, from which it can be seen that the channel is *n*-type doped with a concentration of 6 × 10^14^ cm^−3^. The doping concentration of these two *p*-type areas is 1 × 10^18^ cm^−3^. Two depletion layers are formed between these two *p*-type areas and the *n*-type channel, compressing the conductive area in the channel. The simulated electric field suggests that the built-in field is located in the junction region (Appendix A). According to the band structure simulation results (Figure 2b), the energy band in the *n*-type channel is bent downward, indicating that the built-in field points from the *n*-type channel to the *p*-type area. This means that photogenerated electrons move toward the *n*-type channel, while photogenerated holes migrate to the *p*-type area. Therefore, illumination increases the electron density within the channel compared to the dark state (Figure 2c). In addition, it can also be seen from Figure 2c that light shrinks the depletion layer and expands the conductive area, which is attributed to the modulate effect of the photovoltage [27]. Accordingly, the electron density inside the channel not only increases, but its distribution also expands to both sides of the channel (Figure 2d). As a result, the PVPC outputs a large photocurrent while maintaining a low dark current (Figure 2e).

### 2.3. Effect of Structural Parameters on Performance

Here, we analyze the effect of channel doping concentration on the performance of the PVPC (Figure 3a). As shown in Figure 3b, as the doping concentration increases, the dark current first decreases and then increases, with a minimum of 8.8 nA. The responsivity gradually increases with the increase of doping concentration and then tends to saturation, with a maximum of 3500 A/W. The high responsivity indicates the presence of gain, which can be attributed to the modulation effect of photovoltage. The high dark current at high doping concentration is caused by the narrow depletion layer width (Appendix A). By extracting the electron density in the center of the channel, it can be found that when the doping concentration is low, the depletion layer maintains the electron density at a low value (Figure 3c). However, when the doping concentration is high, the electron density and dark current increase because the channel cannot be effectively depleted.

The variation of responsivity with doping concentration is related to electrostatic potential and dark current. As shown in Figure 3d, the higher the doping concentration, the steeper the electrostatic potential on both sides of the channel, that is, the larger the built-in field. A large built-in field is conducive to the separation of photogenerated electron–hole pairs and the formation of large photovoltage, which enables the channel to be effectively modulated, thereby promoting the photocurrent (*I_ph_*). However, excessive doping concentration increases the dark current (*I_Dark_*) and causes the net photocurrent (*I_Net_* = *I_ph_* − *I_Dark_*) to decrease. Therefore, the responsivity first increases with the increase of doping concentration and then tends to be stable. Based on the simulation results, a doping concentration of about 6 × 10^14^ cm^−3^ is more appropriate to obtain a balance between dark current and responsivity.

Next, the effect of geometric structure on the performance of the PVPC is investigated (Figure 4a). When the doping concentration, channel length (*L_channel_*), and channel width (*W_channel_*) are fixed, the dark current can be suppressed by increasing the doping region length (*L_dope_*) and doping region width (*W_dope_*), that is, increasing the *W_dope_*:*W_channel_* and *L_dope_*:*L_channel_* (Figure 4b). This is because the increase in the size of the doping region introduces more dopants and expands the depletion layer, thereby further depleting the channel. However, unrestrained channel depletion also weakens the photovoltage modulation effect and attenuates the responsivity (Figure 4c). To balance the dark current and responsivity, *W_dope_*:*W_channel_* and *L_dope_*:*L_channel_* can be set to ~0.2 and ~0.3, respectively.

When the doping concentrations *W_dope_*:*W_channel_* and *L_dope_*:*L_channel_* are fixed, the dark current can also be suppressed by reducing the channel width and increasing the channel length (Figure 4d). These results conform to the resistance (*R*) formula [28] (Equation (1)), where *I_Dark_* is the dark current, *ρ* is the resistivity, *S* is the cross-sectional area of the channel, *L* is the channel length, *H* is the channel thickness, *W* is the channel width, *n* is the free electron concentration, *q* is the elementary charge, and *μ* is the mobility. According to Equation (1), it can be seen that the dark current is proportional to *W* and inversely proportional to *L*. In addition, a small channel width is also beneficial for the *p*-*n* junction to deplete free carriers (n) in the channel. However, if the channel width is too small, it becomes difficult for the photovoltage to compress the depletion layer, resulting in decreased responsivity (Figure 4e). Moreover, the channel length should not be too short or too long. When the channel length is too short, the dark current is high. When the channel length is too long, the recombination effect reduces the number of photogenerated carriers. These two factors can lead to a degradation in responsivity. A suitable solution is to design the channel length to be ~8 μm and the channel width to be ~1 μm.
(1)$$I_{Dark}^{-1}\propto R=\rho \frac{L}{S}=\frac{1}{nq\mu }\times \frac{L}{WH}$$

### 2.4. Transient Response and Performance Comparison

Furthermore, the transient response of the PVPC was simulated. As shown in Figure 5a, the rise time and fall time of the PVPC are 2.4 μs and 5.8 μs, respectively, which are comparable to and shorter than those of other low-dimensional material photoconductors, photodiodes, and phototransistors [29,30,31,32,33,34,35,36,37,38,39,40]. Figure 5b,c show the I–V curves of the PVPC and the conventional photoconductor under different light power densities, respectively. The corresponding calculated responsivities of these two types of devices can be found in Figure 5d,e. It should be noted that when the light power is high, the responsivity of the PVPC decreases slightly. This phenomenon can be attributed to the fact that when the depletion layer is almost completely compressed under high light power, it becomes difficult to generate more photovoltage by continuing to increase the light power. Benefitting from the low dark current, the on-off ratio of the PVPC (~368.5) is 277 times higher than that of the conventional photoconductor (~1.33), which is conducive to distinguishing subtle differences in light power. In addition, the PVPC also exhibits higher responsivity compared to the conventional photoconductor. More importantly, the PVPC shows a low equivalent noise power of 2.75 × 10^−15^ W·Hz^−1/2^, underscoring its commendable sensitivity (Appendix A).

## 3. Experimental Section

The energy band, electron density, electrostatic potential, I–V curves, and transient response of the PVPC were obtained using computer-aided design software simulations. To obtain the electronic and hole concentration, Fermi–Dirac statistics was assumed. The equations of Fermi–Dirac statistics are shown below:$$\left\{\begin{array}{c} n=N_{C}F_{1/2}\left(\frac{E_{Fn}-E_{C}}{k_{B}T}\right)\\ p=N_{V}F_{1/2}\left(\frac{E_{V}-E_{Fp}}{k_{B}T}\right) \end{array}\right.$$
where, *n* is the electron concentration, *p* is the hole concentration, *N_C_* is the effective density-of-states of the conduction band, *N_V_* is the effective density-of-states of the valence band, *E_F_n__* is the quasi-Fermi energy for electrons, *E_F_p__* is the quasi-Fermi energy for holes, *E_C_* is the conduction band energy, *E_V_* is the valence band energy, *k_B_* is the Boltzmann constant, and *T* is the temperature.

The conductive current of the PVPC can be obtain by using the following electrostatic and transport equations,
$$\left\{\begin{array}{c} J_{n}=-qn\mu_{n}\nabla E+k_{B}T\mu_{n}n\\ J_{p}=-qp\mu_{p}\nabla E-k_{B}T\mu_{p}p \end{array}\right.$$
where, *J_n_* is the electron current density, *J_p_* is the hole current density, *q* is the elementary charge, *E* is the potential, *μ_n_* is the electron mobility, and *μ_p_* is the hole mobility.

The contact between the electrode and the material was set as Ohmic contact. The temperature was set to 300 K. The doping concentration of channel, channel length, channel width, doping region length, and doping region width were set as variable parameters during the simulation. The thickness of the PVPC was set as 1 μm to ensure sufficient NIR light absorption. NIR light was set to exclusively generate free carriers in the channel region.

## 4. Conclusions

In summary, a PVPC based on a horizontal *p*-*n*-*p* junction has been developed to combine the depletion effect of the *p*-*n* junction and the modulation effect of the photovoltage. In the NIR region (1550 nm), the PVPC simultaneously achieves high responsivity (19 A/W), fast response speed (2.4 μs), and low dark current (9 nA). Further performance improvement can be realized by selecting a suitable doping concentration or optimizing device size. These results provide a promising way for designing photodetectors that balance responsivity, response speed, and dark current, thereby achieving high-performance infrared photodetection.

## Figures and Tables

**Figure 1 nanomaterials-14-01483-f001:**
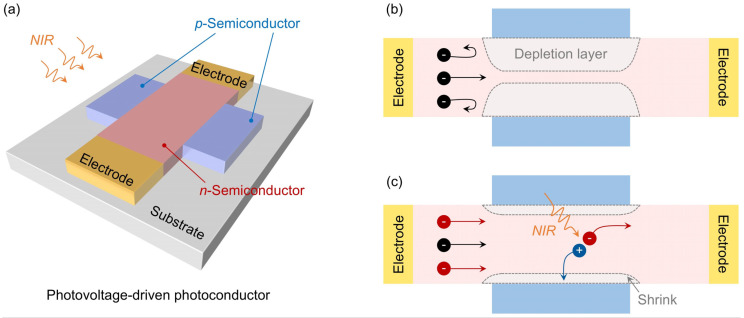
Structure and mechanism of the PVPC. (**a**) Three-dimensional model of the PVPC. (**b**,**c**) Schematic diagrams of the carrier dynamic behavior in the PVPC under darkness and NIR illumination, respectively.

**Figure 2 nanomaterials-14-01483-f002:**
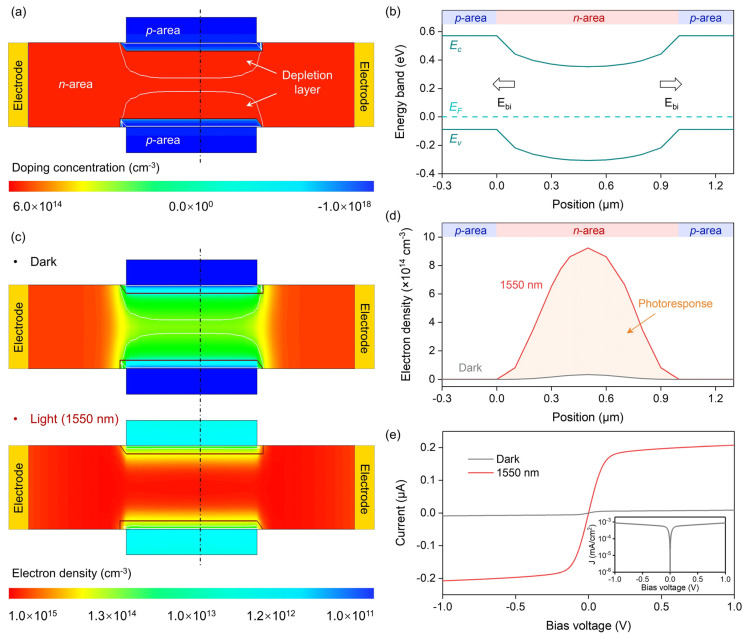
Basic properties of the PVPC. (**a**) Simulated doping concentration profile. The depletion layer is indicated by the white line. (**b**) Energy band along the dashed line in (**a**). (**c**) Simulated electron density in the dark and under 1550 nm illumination. (**d**) Electron density along the dashed line in (**c**). (**e**) Simulated I–V curves in the dark and under 1550 nm illumination. Inset: dark current density (J) plotted in logarithmic form. The power density of 1550 nm illumination is 100 mW/cm^2^. The channel size is 10 μm × 1 μm. The *p*-type doped region size is 4 μm × 0.3 μm.

**Figure 3 nanomaterials-14-01483-f003:**
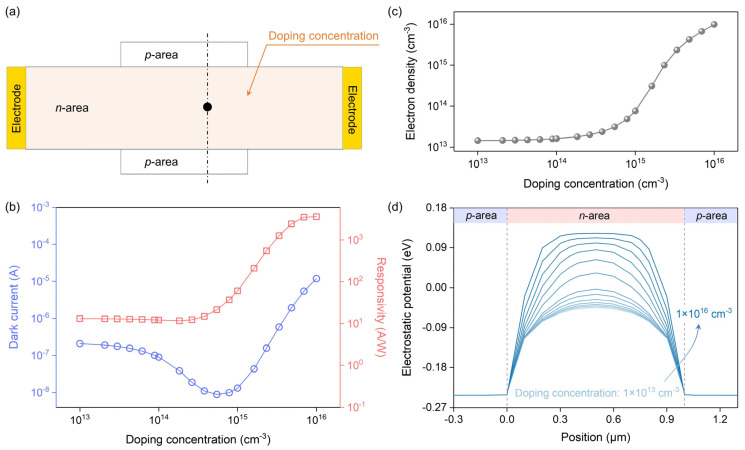
Effect of channel doping concentration on performance. (**a**) Schematic diagram of the PVPC structure. Doping concentration is set as a variable parameter (highlighted by light yellow). (**b**) Doping concentration dependence of the dark current and responsivity. Power density of 1550 nm light is 2 mW/cm^2^. (**c**) Relationship between electron density in the center of the channel (the black dot in (**a**)) and doping concentration. (**d**) Electrostatic potential along the dashed line in (**a**). The electrostatic potential is extracted from the PVPCs with different channel doping concentrations. The channel size is 10 μm × 1 μm. The *p*-type doped region size is 4 μm × 0.3 μm.

**Figure 4 nanomaterials-14-01483-f004:**
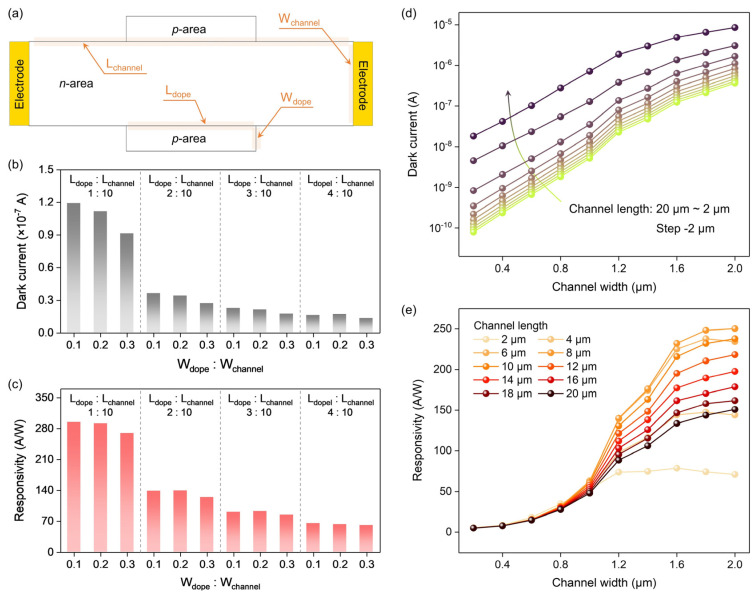
Effect of geometric structure on performance. (**a**) Schematic diagram of the structure of the PVPC. Channel length, channel width, doping region length, and doping region width are set as variable parameters (highlighted by light yellow). (**b**) Relationship between dark current and *W_dope_*:*W_channel_* at different *L_dope_*:*L_channel_*. (**c**) Relationship between responsivity and *W_dope_*:*W_channel_* at different *L_dope_*:*L_channel_*. (**d**) Relationship between dark current and channel width at different channel lengths. (**e**) Relationship between responsivity and channel width at different channel lengths. Doping concentration of channel is 1 × 10^15^ cm^−3^. Power density of 1550 nm illumination is 2 mW/cm^2^. For (**b**,**c**), channel length and channel width are fixed to 10 μm and 1 μm, respectively. For (**d**,**e**), *W_dope_*:*W_channel_* and *L_dope_*:*L_channel_* are fixed to 0.3 and 0.4, respectively.

**Figure 5 nanomaterials-14-01483-f005:**
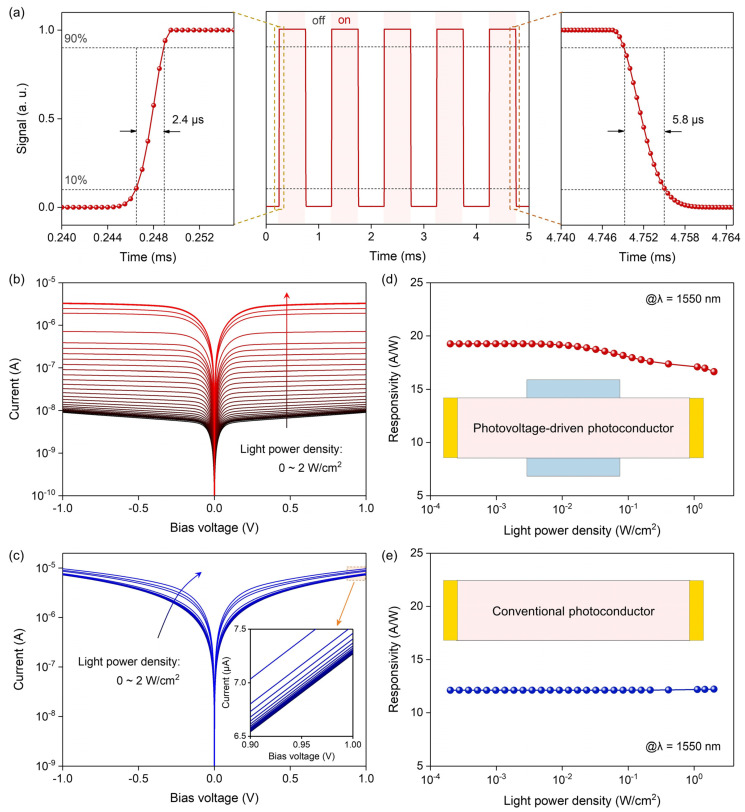
Photodetection performance of the PVPC. (**a**) Simulated transient response. Left panel: enlarged view of rising edge. Middle panel: complete I–T curve. Right panel: enlarged view of falling edge. Power density of 1550 nm illumination is 2 mW/cm^2^. (**b**,**c**) I–V curves of the PVPC and conventional photoconductor under different light power densities. (**d**,**e**) Relationship between responsivity and light power density of the PVPC and conventional photoconductor. The photosensitive area and channel doping concentration of these two types of devices are 1 μm × 10 μm and 5 × 10^14^ cm^−3^, respectively.

## Data Availability

All data needed to evaluate the conclusions in the paper are present in the paper and/or the Appendix A. Additional data related to this paper may be requested from the authors.

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
