# Peer review of "Photovoltage-Driven Photoconductor Based on Horizontal p-n-p Junction"

_nanomaterials, 2024, doi:10.3390/nano14181483_

Round 1

Reviewer 1 Report

Comments and Suggestions for Authors

This manuscript entitled “Photovoltage-driven photoconductor based on horizontal p-n-p junction” by Han et al. presented an interesting device design for developing a high-performance photoelectric device. They used a simulation approach and designed a p-n-p structure to study the effect of doping and channel geometry on the photo-electric performance. Due to the flexibility of the simulation approach, they varied the parameters above to collect the electrostatic parameters of the devices. They tried to correlate them with the photoresponses and responsivity. They obtained extremely high responsivity and decent speed, which seems promising for experimental research to mimic the simulated structures. I would like to recommend acceptance to this study after addressing the comments below.

The title “Photovoltage-drive photoconductor” should be revised, as there is no evidence of photovoltage in the results. The prepared p-n-p channel does not show a photovoltaic effect from the I-V characteristic plots. This should be revised throughout the manuscript.

For consistency, please state intensity in mW/cm2. 0.1 W/cm2 should be 100 mW/cm2. Please correct this throughout the manuscript.

How the electrostatic profiles shown in Figure 3d are obtained is not given.

According to the photoelectric effect, the responsivity value should not exceed 1 A/W. The responsibility value of 3500 A/W should be scrutinized. 

Current value given in A is confusing and don’t account the device area. Please present the current density in units of mA/cm2.

The experimental section provides shallow information, not enough to reproduce the results. It is suggested that the simulation part be expanded with utilized equations and derivatives.

The motivation for considering Ge semiconductors should be clarified in the introduction section.

Why did authors simulate the case of p-n-p structure? Authors should share their views on the p-n-p structure to guide readers in future studies.

In Figure 2, the authors are recommended to profile the electric-field distribution to provide a helicopter view of the electrostatics of the p-n-p structure.

Authors suggested that their proposed scheme provides high-speed responses due to the trade-off between lifetime and photo-gain. It is recommended that the response speed of various photoelectric devices be summarized and compared with their simulated results. Few references are recommended for the comparisons, such as Si Schottky photodiode in Appl. Phys. Lett. 108 (2016) 141904, CZTS-based photodetector in Sensors and Actuators A 314 (2020) 112231, Sb2Se3 based photodetector in ACS Photonics 11 (2024) 1031−1043, inorganic/organic hybrid electrochemical phototransistors in ACS Appl. Mater. Interfaces 13 (2021) 7498–7509, Sb2S3 thin film photoconductive detectors in Adv. Mater. Interfaces 9 (2022) 2101504, photovoltaic array in Small 19 (2023) 2301702, and TiO2/CuI based self-powered UV photodetector in Nanotechnology 33 (2022) 105202.

Comments on the Quality of English Language

Minor editing of English language required.

Reviewer 2 Report

Comments and Suggestions for Authors

The authors deal with a design of photovoltage-driven photoconductor device. The idea is interesting; however, there are still some issue to be fixed or discussed.

#1 Impact of the doping concentration
The doping will not influence only the electron density; however, the charge carrier mobility will be also affected. The authors should discuss influence of the mobility change on the current response.

#2 Dark current dependence
The authors claim that "the increase in doping concentration causes Wn to increase first and then decrease". However, the Eq.(1) has monotonous character for doping concentration change. It is not possible to obtain any local maxima/minima. Authors should explain it clearly.

#3 Photodetector parameters
Since authors evaluated photodetector devices, other parameters, such as noise equivalent power and detectivity, should be estimated. 

As a result, I would like to support the manuscript if above-mentioned issues will be fixed. Hence, I suggest only minor changes.

Comments on the Quality of English Language

No comments to English level.
